# Association of Homozygous *PROP1* Mutation in a Saudi Family with Combined Pituitary Hormone Deficiency

**DOI:** 10.3390/medicina59030474

**Published:** 2023-02-27

**Authors:** Ahmed M. Almatrafi, Ali M. Hibshi, Sulman Basit

**Affiliations:** 1Department of Biology, College of Science, Centre for Genetics and Inherited Diseases, Taibah University Al Madinah, Al Munawarah 42353, Saudi Arabia; 2Department of Obstetrics & Gynecology, King Sulman Medical City-Madinah Maternity and Children Hospital, Almadinah Almunawwarah 42319, Saudi Arabia; 3Department of Biochemistry, College of Medicine, Centre for Genetics and Inherited Diseases, Taibah University Al Madinah, Al Munawarah 42353, Saudi Arabia

**Keywords:** combined pituitary hormone deficiency, *PROP1*, novel sequence variant, whole-exome sequencing

## Abstract

*Background and Objectives*: Combined pituitary hormone deficiency (CPHD) is a rare heterogeneous disease. It is characterized by the deficiency of growth hormone (GH) and shortage of at least one or more other hormones of the pituitary gland including thyroid-stimulating hormone (TSH), luteinizing hormone (LH), follicle-stimulating hormone (FSH), and prolactin. Rare pathogenic variants in nearly 30 genes have been identified as an underlying cause of CPHD pathogenicity. Among these genes, paired-like homeobox 1 (*PROP1*) has been reported to be the most common cause of CPHD. *Materials and Methods*: In the present study, we investigated a large family of Saudi origin with three adult sisters suffering from short stature in combination of secondary amenorrhea. *Results*: Whole-exome sequencing followed by Sanger sequencing shows a homozygous missense variant (NM_006261.5; c.211C > T; p.R71C) in the *PROP1* gene segregating with the disease phenotype within the family. In silico analysis studies show that this variant is highly conserved among several orthologues and is predicted as likely pathogenic using various bioinformatics tools. *Conclusions*: Our finding presents the first Saudi familial case of autosomal recessive form of CPHD caused by the *PROP1* variant.

## 1. Introduction

Combined pituitary hormone deficiency (CPHD) is a phenotypically variable and genetically heterogeneous disorder. The disease manifestations can occur anywhere between births to puberty. It involves impaired secretion of growth hormone (GH) in association with shortages of other posterior and/or anterior pituitary hormones [1]. Patients with CPHD may exhibit several clinical symptoms such as hypothyroidism, hypocortisolism, short stature, and impaired sexual maturation at the age of puberty. In addition, with or without extra-pituitary phenotypes, some cases might develop hypoplasia of the optic nerve and midline forebrain abnormalities [2,3]. It is reported that 45% of patients with isolated GH deficiency (IGHD), which involves deficiency of GH only, may also develop CPHD over time, particularly a severe type [4].

The frequency of CPHD is estimated to be 1 in 8000 individuals globally [5]. The majority of incidence is caused by non-genetic factors such as brain surgery, infection, irradiation, tumor, chronic poisoning by heavy metals, trauma, and autoimmune diseases. In addition, the condition can be caused by genetic defect in genes associated with the development and maintenance of the anterior part of the pituitary gland [5,6,7,8]. These genetic defects mainly occur spoIt must be radically, however, several familial forms have been reported in different inheritance patterns including autosomal dominant, recessive, and X-linked recessive [9,10]. It is estimated that 5–30% of CPHD cases are familial cases [3,11].

So far, approximately 30 genes have been demonstrated to be implicated in CPHD pathogenesis. Mutations in *POU1F1*, *LHX3*, *HESX1*, *LHX4*, *SOX3*, *GLI2,* and *PROP1* are the most frequent causes of CPHD [12]. Most of these genes encode transcription factors involved in normal embryonic pituitary cell differentiation and pituitary organogenesis [13]. *PROP1* mutations are the most commonly detected cause of familial CPHD cases, accounting for 30–50%. Nearly 10% of these cases have an affected first-degree relative. Although *PROP1* mutations are rare in sporadic CPHD cases, accounting for nearly 6.7% [1,3,14]. 

The deficiency in Prop1 was initially found in Ames mice leading to dwarf mice and impairment of the level of growth hormones (GH), prolactin (PRL), and/or thyroid-stimulating hormone (TSH) [15]. In humans, the clinical features associated with *PROP1* gene mutations are extremely variable between individuals from the same family as well as between individuals from different pedigrees in many respects, including the time of disease onset and severity of the disease, function of adrenal gland, onset of puberty signs, and the final height of untreated patients [16]. 

In the present study, we describe a clinical and genetic characterization of a Saudi family with three sisters exhibiting features of short stature, secondary amenorrhea, and deficiency in pituitary hormones. A homozygous variant in the *PROP1* gene was identified in all three affected individuals. This is the first report of a Saudi familial case of an autosomal recessive form of CPHD caused by the *PROP1* variant.

## 2. Materials and Methods

### 2.1. Family Recruitment and Ethical Approval 

A four-generation family of a Saudi origin including three adult sisters showing symptoms of secondary amenorrhea with short stature were clinically examined at reproductive endocrinology and infertility clinic at Madina Maternity and Children Hospital (MMCH), Medina. History of the disease was obtained from elders of the family and a pedigree was drawn to assess the inheritance of the disorder (Figure 1). The study research was approved by the institutional review board of Taibah University (TUCDREC/27032021) and MMCH (IRB147-2021). Patients signed the written informed and disclosure consent form after receiving an explanation of the study aim to family members in Arabic language. All the experiments in the current research were conducted in accordance with the guidelines of the ethical committee. 

### 2.2. DNA Extraction and Quantification 

A total of 3 mL of peripheral blood samples was obtained from each participant of the family including three affected (IV-1, IV-5, and IV-6) and three unaffected (IV-2, IV-3, and IV-4), individuals as well as from both parents (III-1 and III-2) in ethylenediaminetetraacetic acid tubes for genetic analysis. Genomic DNA was extracted using the QiaAmp DNA mini kit (Qiagen cat no.51306) according to manufacturer’s protocol. The integrity and purity of DNA was measured using Qubit fluorometer and Nanodrop-1000 spectrophotometer (Thermo Fisher Scientific, Waltham, MA, USA).

### 2.3. Next-Generation Sequencing

DNA samples of two affected sisters (IV-1 and IV-6) were whole-exome sequenced (WES) and data were analyzed according to the protocol described elsewhere [17]. Briefly, the SureSelect Kit (Agilent, Santa Clara, CA, USA) was used for preparation of DNA libraries and the Illumina Hiseq2000 instrument was used for generating sequence reads (Illumina, San Diego, CA, USA). The online Illumina BaseSpace analysis tool (https://basespace.illumina.com, accessed on 13 November 2022) was used for standard filtration of variant calling files (VCF) of the two affected participants. Based on family pedigree and consanguineous marriages, it is clear that the disorder depicted an autosomal recessive inheritance pattern. Therefore, only common homozygous and compound heterozygous variants among the two affected participants were filtered. Variants were classified based on the guidelines of ACMG [18]. 

### 2.4. Sanger Sequencing Validation 

After filtration and variant prioritization, specific primers were designed using the Primer 3 tool (http://primer3.ut.ee/ accessed on 15 October 2022) to validate mutated variants. Ensembl genome browser (https://m.ensembl.org/ accessed on 15 October 2022) was used for downloading the reference sequence of genes of interest. Bidirectional Sanger sequencing was performed for all identified variants using basic protocol as previously mentioned. The sequenced reads were aligned with the reference sequence using the BIOEDIT software version 6.0.7 (https://bioedit.software.informer.com, accessed on 13 November 2022) for variant confirmation. 

### 2.5. In Silico Analysis

In silico analysis was carried out to predict the pathogenicity effect of identified variants on overall protein structures. Different online tools were used such as MutationTaster (http://www.mutationtaster.org/; accessed on 20 November 2022), Mutation Assessor (http://mutationassessor.org/; accessed on 20 November 2022), VarSome (https://varsome.com/; accessed on 20 November 2022), SIFT (http://sift.bii.a-star.edu.sg/; accessed on 8 November 2022), FATHMM (http://fanthmm.biocompute.org.uk/; accessed on 20 November 2022), PolyPhen-2 (http://genetics.bwh.harvard.edu/pph2/; accessed on 8 November 2022), CADD (https://cadd.gs.washington.edu/; accessed on 8 November 2022), and PredictProtein (https://predictprotein.org/; accessed on 20 November 2022). In addition, the frequency of discovered variants among the general population was investigated through various online databases including EVS (http://evs.gs.washington.edu/EVS/; accessed on 20 November 2022), gnomAD (http://gnomad.broadinstitute.org/; accessed on 20 November 2022), 1000 genomes (http://www.internationalgenome.org/; accessed on 20 November 2022), ExAC browser (http://exac.broadinstitute.org/; accessed on 20 November 2022), and exomes data of 125 healthy Saudi individuals. NCBI HomoloGene (http://www.ncbi.nlm.gov/homologene/, accessed on 13 November 2022) was used for identification of conservation of p. Arg71Cys amino acid in various PROP1 orthologs.

UNIPROT (https://www.uniprot.org (accessed on 28 December 2022)) was used for mapping of the identified mutation to the PROP1 domain. STRING (version 11.5) website (https://string-db.org/; accessed on 1 January 2023) was used for identification interactions network.

## 3. Results

### 3.1. Clinical Evaluation 

All three affected sisters (IV-1, IV-5, and IV-6) shared similar clinical features of short stature and irregular menstrual cycle. Both parents and the three other healthy siblings had normal growth, normal weight, and normal development of secondary sexual traits, and did not display any abnormal features. The heights of father and mother were 159 cm and 149 cm, respectively. The height range of healthy siblings was between 151 to 163 cm. 

Medical history of the three affected sisters showed that they were born after uneventful pregnancies and normal vaginal deliveries, with more than 2.5 kg birth weight. During the early childhood periods, all three patients showed regular developmental milestones with no significant medical complaints. 

At the age of 13, the parents observed the growth failure of the first daughter (IV-1) during her admission in middle school. She was referred to an endocrinologist for clinical evaluation. Investigations revealed profound GH deficiency. GH injection and oral contraceptive pills were administered for one year. Follow up showed a 3 cm increase in her height. However, at the age of 16, a menarche with irregular menstrual cycle was observed, but she was, overall, satisfied with the medication. At the age of 34 years, the patient (IV-1) was referred to the endocrinology clinic at MMCH due to secondary amenorrhea with short stature. The hormonal analysis results showed normal follicle-stimulating hormone (FSH) 3.2 lU/L (N: 1.5–12.4 lU/L) and luteinizing hormone (LH): 6.06 lU/L (N: 5–25 lU/L). However, she had high thyroid-stimulating hormone (TSH) 4.4 m lU/L (N: 0.4–2.5 m lU/L) and prolactin (PRL) 110mU/L (N: <25 μg/L). Clinical examination showed that she had normal breast development Tanner stage V, and normal pubic and axillary hair.

Her affected sibling (IV-5), aged 28 years, complained of an irregular menstrual cycle and short stature. Her height was 137 cm and weight 36.8 kg with normal mental function. She completed her bachelor’s degree. At the age of 19, she had spontaneous menarche for two years and at 21 years of age, she developed amenorrhea for one year. The hormonal profile of IV-5 is presented in Table 1. The patient had normal breast development Tanner stage V, and normal pubic and axillary hair. Ultrasonography of the ovaries showed normal echogenicity ovaries with no suspicious cystic or solid lesions. The size of the right ovary was 3.8 cm^3^ and the left, 2.2 cm^3^.

The third affected female (IV-6) is currently 23 years old, displaying an irregular menstrual cycle and short stature. Her height and weight is 137 cm and 43.9 kg, respectively. She induced menarche at the age of 23 years of age. The patient had Tanner stage III with normal pubic hair, and she had no axillary hair. The hormonal profile of IV-6 is presented in Table 1. 

### 3.2. A Homozygous Variant in the PROP1 Gene Was Discovered

High quality exome sequence reads for individuals (IV-1) and (IV-6) were obtained with more than 100 x coverages. The analysis of the exome data, including annotation of variants, filtration, and prioritization, reveals a homozygous variant C-to-T transition (c.211C > T) in the *PROP1* gene (Figure 2). This homozygous variant in *PROP1* was predicted as the most possible pathogenic variant among the family based on the previous studies of the involvement of *PROP1* variants in developing CPHD syndrome (Table 2). 

This missense homozygous variant (c.211C > T) in the PROP1 gene is predicted to change amino acid arginine to cysteine (p.R71C) at codon 71 in exon 2 with a CADD score of 24.6. This single nucleotide change is in the N-terminal region of homeodomain. No other pathogenic variants were detected in known CPHD genes.

### 3.3. Validation of Variants with Sanger Sequencing

The exon 2 of *PROP1* gene was sequenced in all available family members by the Sanger approach. The analysis of Sanger reads shows recessively inheritance of variant in a pedigree. Both parents and the three heathy siblings are heterozygous for the mutant allele, whereas all the three affected individuals are homozygous for the variant (Figure 3A).

### 3.4. Evolutionary Conservation of PROP1 Amino Acid at Position 71

Multiple sequence alignment was performed to identify the evolution of PROP1 protein at position 71 of the amino acid. Sequence alignment shows that this position is highly conserved in various species throughout evolution. The conservation of the arginine acid at this location suggests the importance of this amino acid to the protein structure and function (Figure 3C).

### 3.5. Protein Network Interaction Analysis of PROP1

Protein–protein interaction (PPI) was used to investigate potential interactions between PROP1 and other associated proteins. Our data in Figure 4 show the network containing 11 nodes and 35 edges with PPI enrichment; *p*-value 0.00000029. Most notably, PROP1 interacts with different proteins involved in vital biological functions in the pituitary, mammary, and thyroid glands organogenesis including *POU1F1*, *SOX3*, *POMC*, *TSHB*, *PRL*, *TBX19*, *GHRHR*, and *GHRH*. Also, the network shows the strong association of PROP1 with other important protein such as furin and catenin beta-1 (CTNNB1).

## 4. Discussion

Mutations in transcription factor genes that encode proteins for hypothalamic and pituitary gland function, or differentiation in development can lead to congenital CPHD. CPHD is a rare medical condition that is associated with a wide spectrum of defects including growth abnormality and dysfunction of anterior pituitary causing hyperplasia or hypoplasia [19]. The current study presents phenotypic features and genotypic analysis of a consanguineous Saudi family with CPHD segregating among the family members in an autosomal recessive pattern. A homozygous missense variant c.211C > T: p.(Arg71Cys) in exon 2 (NM_006261.5) of PROP1 was identified for the CPHD phenotype in the family. This variant has been described on the Varsome website as a variant of uncertain significance and had not been previously reported on the ClinVar website. However, an earlier study revealed two novel compound heterozygous mutations in (p.(Arg71Cys)/p.(Arg71His)) of the homeodomain of the first alpha helix of the *PROP1* gene similarly affecting the amino acid in two pre-pubertal siblings with CPHD [20].

Familial CPHD due to an inactivating mutation of the *PROP1* gene was first reported in 1998. This transcriptional factor was reported previously to play a crucial role in the proper development of somatotrophs, lactotrophs, thyrotrophs, and gonadotrophs [21]. It is composed of three exons that encode 226 amino acids paired-like homeodomains, and it is located in the long arm of chromosome 5q35.3 [22].

In humans, PROP1 deficiency is most frequently associated with defects in GH, FSH, PRL, TSH, and LH; however, deficiency of ACTH is rarely observed. The hormonal profiles of the three affected sisters are different even though they share a similar missense mutation. Interestingly, patient (IV-6) has LH deficiency while the other affected sisters, IV-1 and IV-5, show abnormal TSH results. All three patients have normal FSH and abnormal PRL results. For analysis of free thyroxine (FT4), patients IV-5 and IV-6 have high results of FT4. However, this hormonal test was not performed for patient IV-1. Clinically, two affected sisters, IV-1 and IV-5, are at Tanner stage V with normal pubic and axillary hair, whereas, IV-6 is at Tanner stage III with normal pubic hair and no axillary hair.

This result is consistent with a previous report that individuals with mutations in *PROP1* might exhibit different phenotype/genotype correlations. A large study including 73 affected patients with CPHD from 36 families showed high variability in the clinical features and pituitary hormonal (GH, LH, FSH, and TSH) deficiencies that reduced with age progressively, following a distinct pattern and time scale in each CPHD patient [23].

The rate of *PROP1* mutations differ significantly between ethnicities. In the previously reported CPHD cases, the most frequent PROP1 mutations among Latin American, Eastern European, and Portuguese patients were c.301_302delAG and c.150delA of exon 2 [12]. These two mutations were found in 90% of CPHD cases. On the other hand, Moalla et al. (2022) recently reported two mutations (p.(Gln114Ter) and p.(Arg73Cys)) in Tunisian families with non-syndromic CPHD [24]. The p.(Arg73Cys) mutation was previously reported as the most common PROP1 mutation among CPHD patients from North Africa [25,26,27]. However, the mutation in p.(Arg71Cys) was described once in a non-consanguineous Italian family as compound heterozygotes with p.(Arg71His) resulting from different clinical features of both affected siblings associated with deficiency in GH and TSH [20]. In this study, we identified no phenotypic effect of heterozygotes mutation p. (Arg71Cys) in PROP1 in either the parents or healthy siblings.

PROP1 has both transcriptional activation and DNA-binding ability [1]. Our amino acid alignment result shows p.(Arg71Cys) present in the paired homeo DNA-binding domain, which is highly conserved among other species including bovines, pigs, monkeys, dogs, gorillas, and Gelada baboons. PPI analysis suggests the crucial role of PROP1, which interacts with other families including the Pit-Oct-Unc (POU) transcription factor family; proprotein convertase subtilisin/kexin family; glucagon family; pro-opiomelanocortin; met-enkephalin family, growth hormone family, and the Wnt signaling pathway.

Finally, our patients currently are under estrogen and progestin hormone replacement therapeutics, using ethinylestradiol and norethindrone in combination with norimin to treat the secondary amenorrhea. In addition, patients are using both calcium carbonate and vitamin D3 to prevent low blood calcium and vitamin D deficiency. However, this study has some limitations, including not performing other hormonal analysis such as GH, ACTH, hypercholesterolemia, and insulin-like growth factor 1, as it was not available in the local hospital.

In conclusion, this new likely pathogenic missense mutation (c.211C > T; p.(Arg71Cys)) in the *PROP1* gene expands the phenotypic and genetic spectrum of CPHD syndrome. To the best of our knowledge, this is the first report of PROP1 mutation causing CPHD among the Saudi population. The study recommends genetic screening of patients who display clinical phenotypes of short stature with pituitary hormone deficiency to provide early diagnosis of CPHD for adequate follow-up and to avoid any clinical complications in future.

## Figures and Tables

**Figure 1 medicina-59-00474-f001:**
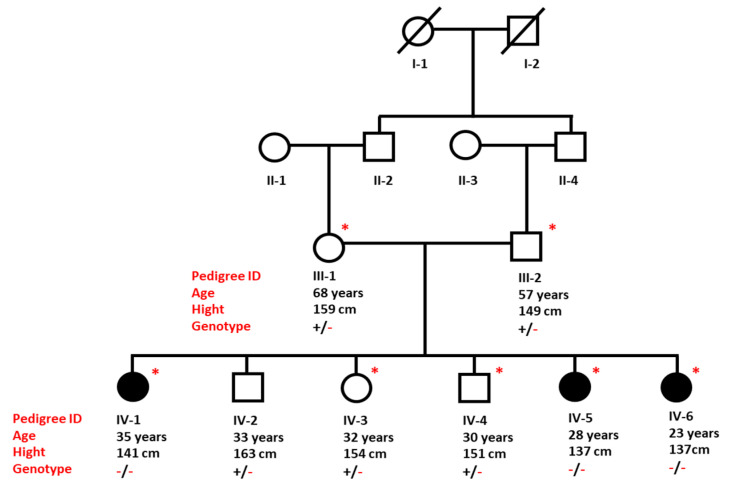
Pedigree of a Saudi family presenting inheritance of CPHD in an autosomal recessive pattern. Individuals marked with asterisks (*) are available for clinical and genetic analysis.

**Figure 2 medicina-59-00474-f002:**
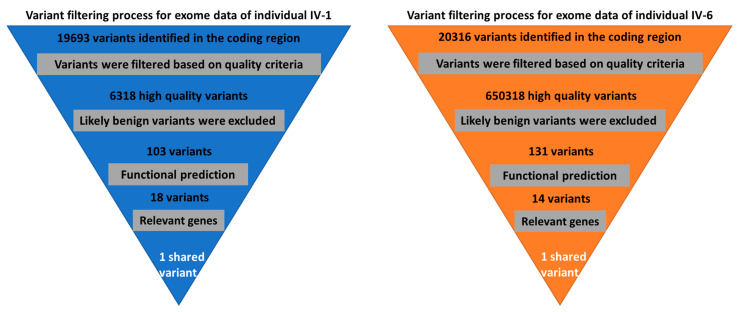
Variant filtration scheme for the exome data of two affected individuals. A single shared homozygous variant has been detected using this scheme.

**Figure 3 medicina-59-00474-f003:**
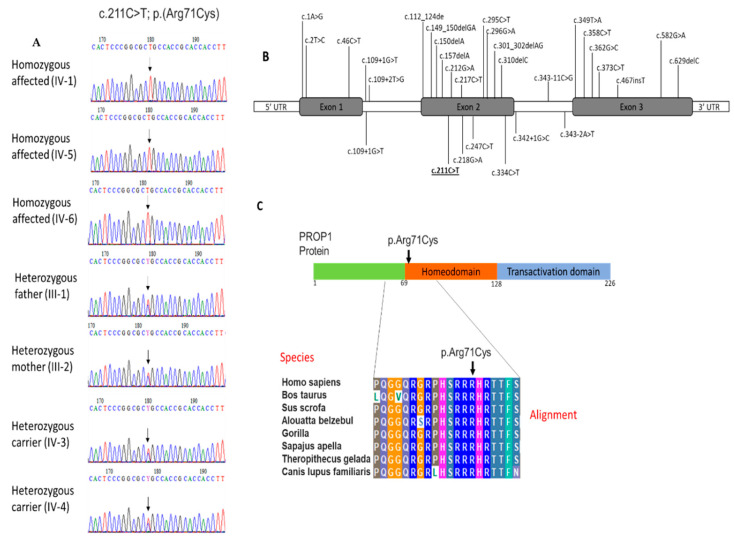
(**A**) Sequence analysis of missense variant in *PROP1* (c.211C > T) in the affected individuals (IV-1), (IV-5), (IV-6) and heterozygous carriers father (III-1) and mother (III-2), as well as other phenotypically normal sister (IV-3) and brother (IV-4); (**B**) location of up-to-date variants of *PROP1* associated with CPHD, including our findings. Result of this study is underlined; (**C**) PROP1 protein domain showing homeodomain region and multiple sequence alignment of a partial amino acid sequence of a human PROP1 protein with its orthologues in other species. The amino acid arginine at position 71 is highly conserved in different species.

**Figure 4 medicina-59-00474-f004:**
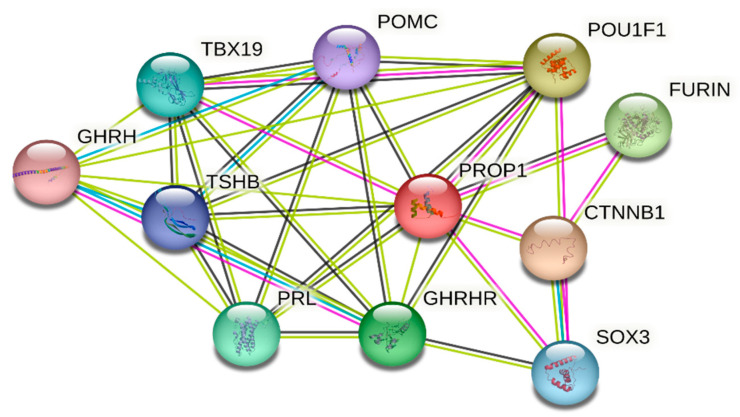
PROP1 protein–protein interaction network analysis using STRING database with confident sore 0.4. Strong association between PROP1 and other pituitary protein including thyroid-stimulating hormone POU1F1 and TSHB; short-stature-protein-associated POU1F1, GHRHR, SOX3, and CTNNB1; amenorrhea and infertility POU1F1 and SOX3 proteins.

**Table 1 medicina-59-00474-t001:** Common clinical and hormonal findings of affected individuals.

Parameters	IV-1	IV-5	IV-6	Reference Range
Age (years)	34	28	23	
Height (cm)	141	137	137	
Weight (Kg)	53	36.8	43.9	
Laboratory hormone results				
FSH	3.2	9.5	2.7	1.5–12.4 lU/L
LH	6.06	8.04	1.1	5–25 mIU/mL
TSH	4.4	4.09	1.4	0.4–2.5m lU/L
FT4	--	13.3	9.3	0.7–1.9 ng/dL
Prolactin (PRL)	110 mU/L	306	127	<25 μg/L
TESTO	--	0.0208	0.025	2.80–8.00 ng/mL
Estradiol 2	--	54.75	--	12–166 pg/mL
Estradiol 3	--	--	5.00	5.10–2950 pg/mL
Clinical features				
Short statue	+ve	+ve	+ve	
Age of induced menarche (years)	16	19	23	
Amenorrhea	+ve	+ve	+ve	
Frontal bossing	+ve	+ve	+ve	
Breast development	Tanner stage V	Tanner stage V	Tanner stage III	
Public hair	Normal	Normal	Normal	
Axillary hair	Normal	Normal	Absent	
High-pitched voice	+ve	+ve	+ve	
Hypothyroidism	−ve	−ve	−ve	
Wrinkled skin	−ve	−ve	−ve	
Dry skin	−ve	−ve	−ve	

**Table 2 medicina-59-00474-t002:** Common pathogenic variant identified by WES and their interpretation with several bioinformatic analysis tools.

Gene	Nucleotide Variant	Protein Variant	Zygosity	gnomAD Exome Freq	ACMG Classification	Prediction Tools					
						SIFT	PolyPhen2	Mutation Taster	CADD	Mutation Assessor	FATHMM
*PROP1*	c.211C > T	p. R71C	Homo	0.0000199	VUS (BP4)	P	LP	DC	24.6	P	VUS

Abbreviations: gnomAD = Genome Aggregation Database, ACMG = American College of Medical Genetics, SIFT = sorting intolerant from tolerant, Polyphen2 = polymorphism phenotyping v2, CADD = combined annotation-dependent depletion, FATHMM = functional analysis through hidden Markov models, VUS = variant of uncertain significance, P = pathogenic, LP = likely pathogenic, DC = disease-causing.

## Data Availability

Data available on request from the authors.

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
