# Peer review of "Association of Homozygous PROP1 Mutation in a Saudi Family with Combined Pituitary Hormone Deficiency"

_medicina, 2023, doi:10.3390/medicina59030474_

Round 1
Reviewer 1 Report
In this article, the authors identified a homozygous missense variant R71C in PROP1 in a Saudi family with 3 affected individuals. The individuals have short stature and secondary amenorrhea. This represents a novel mutation in PROP1 that causes combined pituitary hormone deficiency (CPDH).
The genetics and clinical investigation are sound. I only have minor comments. The article will need some copy-editing to improve the quality of the English. The figures also need to be of higher resolution for publication.
Specific comments as follows:
Figure 2a. Sanger sequencing traces of all the affected individuals should be shown. The authors can also mark in the pedigree in Figure 1 which individuals were sequenced, and their respective PROP1 genotypes.
Figure 2b/c. It is useful to have a diagram of the protein domains in PROP1 so that the readers can understand where the homeodomain is and where the variant is with respect to the homeodomain.
Line 253. “These two mutations were found in 90% of these populations.” This is rather misleading and should be changed to found in 90% of CPHD cases.
The authors should include a filtering chart/ table of the number of common homozygous and compound het variants in the two affected individuals and how that was filtered down to PROP1.
Author Response
Response to Reviewers
[Cover Letter]
Dear Editor,
We appreciate you and the reviewers for your precious time in reviewing our paper and adding valuable comments. It was your valuable and insightful comments that led to possible improvements in the current version. The authors have carefully considered the comments and tried our best to address almost all comments of the reviewers. The manuscript has been revised thoroughly and we hope the manuscript after careful revisions meet your high standards. The authors welcome further constructive comments if any. Below we provide the point-by-point responses. All changes made in the text have been highlighted.
Sincerely,
Comments and Suggestions for Authors
In this article, the authors identified a homozygous missense variant R71C in PROP1 in a Saudi family with 3 affected individuals. The individuals have short stature and secondary amenorrhea. This represents a novel mutation in PROP1 that causes combined pituitary hormone deficiency (CPDH).
The genetics and clinical investigation are sound. I only have minor comments.
Comment: The article will need some copy-editing to improve the quality of the English.
Response: Extensive English editing has been carried out
Comment: The figures also need to be of higher resolution for publication.
Response: I changed figure 1.
Specific comments as follows:
Comment: Figure 2a. Sanger sequencing traces of all the affected individuals should be shown. The authors can also mark in the pedigree in Figure 1 which individuals were sequenced, and their respective PROP1 genotypes.
Response: Individuals available for this study have been marked. Genotypes have been shown below each individual.
Comment: Figure 2b/c. It is useful to have a diagram of the protein domains in PROP1 so that the readers can understand where the homeodomain is and where the variant is with respect to the homeodomain.
Response: Figure with protein domains marked with mutation sites have been added.
Comment: Line 253. “These two mutations were found in 90% of these populations.” This is rather misleading and should be changed to found in 90% of CPHD cases.
Response: Rectified
Comment: The authors should include a filtering chart/ table of the number of common homozygous and compound het variants in the two affected individuals and how that was filtered down to PROP1.
Response: A chart showing variant filtration scheme has been added.

Reviewer 2 Report
The manuscript "Association of homozygous PROP1 mutation in a Saudi family with combined pituitary hormone deficiency" by Almatrafi et al. presents a family with a consanguineous line and ultimately PROP1 mutation in the fourth generation of this family with three homozygous female patients presenting with diverse clinical manifestations. The manuscript needs extensive copy-editing: there are a lot of spelling mistakes ("statue" and "Novermber" to name two examples), some unusual wording, and awkward syntax. However, the manuscript is still easily understandable.
The unique feature of this case report is the novel mutation presented therein, which is very interesting. However, the clinical presentation of the carriers is not adequate. One major limitation is the lack of assessment of other pituitary hormones; in my opinion this limitation is recognized highly enough in the discussion - it is briefly mentioned in a subclause. This must be amended. Furthermore, the TSH value of the patients is not really indicative of anything in these cases - what do the authors want to convey with this information? Is there thyroid pathology? If so, have ultrasound examinations been performed? Antibody testing? Equally, prolactin levels are normal in the patients and there is a difference in units that is not respected (it says in the text: "Prolactin (PRL) 110mU/L (N: <25 ug/L)" 110 mU/L is approximately 5.2 ug/L, so a perfectly normal level). We have short stature as a common denominator and differently phenotypic sexual development (which per se is not indicative of pituitary pathology). So, overall, the presentation of clinical data and its interpretation needs to be revised thoroughly.
Minor aspects:
* poor resolution of figures
* poor quality of fig. 1
Summary:
This is an interesting case due to a novel mutation of PROP1 with a solid base of the manuscript, but poor execution (language, clinical presentation) and therefore needs major revision before being acceptable for publication.
Author Response
Response to Reviewers
[Cover Letter]
Dear Editor,
We appreciate you and the reviewers for your precious time in reviewing our paper and adding valuable comments. It was your valuable and insightful comments that led to possible improvements in the current version. The authors have carefully considered the comments and tried our best to address almost all comments of the reviewers. The manuscript has been revised thoroughly and we hope the manuscript after careful revisions meet your high standards. The authors welcome further constructive comments if any. Below we provide the point-by-point responses. All changes made in the text have been highlighted.
Sincerely,
Comments and Suggestions for Authors
The manuscript "Association of homozygous PROP1 mutation in a Saudi family with combined pituitary hormone deficiency" by Almatrafi et al. presents a family with a consanguineous line and ultimately PROP1 mutation in the fourth generation of this family with three homozygous female patients presenting with diverse clinical manifestations.
Comment: The manuscript needs extensive copy-editing: there are a lot of spelling mistakes ("statue" and "Novermber" to name two examples), some unusual wording, and awkward syntax. However, the manuscript is still easily understandable.
Response: Extensive English editing has been carried out. Grammatical mistakes and typographic have been removed.
Comment: The unique feature of this case report is the novel mutation presented therein, which is very interesting.
Response: Thank you very much for appreciating the strength of this manuscript.
Comment: However, the clinical presentation of the carriers is not adequate.
Response: Clinical features have been revised. Three carriers including two males and one female sib showed normal clinical features and normal sexual development with no complain. Moreover, both parents showed normal developmental milestones. Hormone analysis was not performed for carriers.
Comment: One major limitation is the lack of assessment of other pituitary hormones; in my opinion this limitation is recognized highly enough in the discussion - it is briefly mentioned in a subclause. This must be amended.
Response: We have recruited these patients after the onset of puberty. They visited and consulted the clinician at adulthood age complaining from secondary amenorrhea and growth statues, so, some pituitary hormones such as GH, ACTH were not assessed.
Comment: Furthermore, the TSH value of the patients is not really indicative of anything in these cases - what do the authors want to convey with this information? Is there thyroid pathology? If so, have ultrasound examinations been performed? Antibody testing? Equally, prolactin levels are normal in the patients and there is a difference in units that is not respected (it says in the text: "Prolactin (PRL) 110mU/L (N: <25 ug/L)" 110 mU/L is approximately 5.2 ug/L, so a perfectly normal level). We have short stature as a common denominator and differently phenotypic sexual development (which per se is not indicative of pituitary pathology). So, overall, the presentation of clinical data and its interpretation needs to be revised thoroughly.
Response: Yes, TSH value is not indicative for secretion shortages, therefore, ultrasound examination and antibody tests were not done.
These three sisters showed obvious short status at least (15 cm) shorter than the carries brothers and one sister.
Minor aspects:
Comment:* poor resolution of figures
Response: Resolution of figures have been improved.
Comment: * poor quality of fig. 1
Response: figure quality enhanced.
Summary:
Comment: This is an interesting case due to a novel mutation of PROP1 with a solid base of the manuscript, but poor execution (language, clinical presentation) and therefore needs major revision before being acceptable for publication.
Response: We have thoroughly revised the manuscript. English language has also been edited. Clinical data has been revised.
